# Strengthening capacity in hospitals to reduce perinatal morbidity and mortality through a codesigned intervention package: protocol for a realist evaluation as part of a stepped-wedge trial of the Action Leveraging Evidence to Reduce perinatal morTality and morbidity (ALERT) in sub-Saharan Africa project

Ibukun-Oluwa Omolade Abejirinde [1,2] Virginia Castellano Pleguezuelo,[3] Lenka Benova,[3] Jean-Paul Dossou,[4] Claudia Hanson [5] Christelle Boyi Metogni,[4] Samuel Meja,[6] D A Mkoka,[7] Gertrude Namazzi,[8] Kristi Sidney,[9] Bruno Marchal [3] The ALERT Study Team

For numbered affiliations see end of article.

**Correspondence to**
Bruno Marchal;
BMarchal@itg.be

## Strengths and limitations of this study

► Realist evaluation enhances the explanatory power of process evaluations by providing an analytical approach that starts from an intervention theory and explains the outcome in terms of a configuration of intervention, context, actors and mechanisms of change.
► The approach furthermore allows testing and developing the theoretical basis of implementation and scaling up of effective interventions.
► Study limitations include social desirability bias and potential risks presented by cross-cultural differences, power imbalances and western-dominated theories in realist research.

## ABSTRACT

**Introduction** Despite a strong evidence base for developing interventions to reduce child mortality and morbidity related to pregnancy and delivery, major knowledge–implementation gaps remain. The Action Leveraging Evidence to Reduce perinatal morTality and morbidity (ALERT) in sub-Saharan Africa project aims to overcome these gaps through strengthening the capacity of multidisciplinary teams that provide maternity care. The intervention includes competency-based midwife training, community engagement for study design, mentoring and quality improvement cycles. The realist process evaluation of ALERT aims at identifying and testing the causal pathway through which the intervention achieves its impact.

**Methods and analysis** This realist process evaluation complements the effectiveness evaluation and the economic evaluation of the ALERT intervention. Following the realist evaluation cycle, we will first elicit the initial programme theory on the basis of the ALERT theory of change, a review of the evidence on adoption and diffusion of innovations and the perspectives of the stakeholders. Second, we will use a multiple embedded case study design to empirically test the initial programme theory in two hospitals in each of the four intervention countries. Qualitative and quantitative data will be collected, using in-depth interviews with hospital staff and mothers, observations, patient exit interviews and (hospital) document reviews. Analysis will be guided by the Intervention-Actors-Context-Mechanism-Outcome configuration heuristic. We will use thematic coding to analyse the qualitative data. The quantitative data will be analysed descriptively and integrated in the analysis using a retroductive approach. Each case study will end with a refined programme theory (in-case analysis). Third, we will carry out a cross-case comparison within and between the four countries. Comparison between study countries should enable identifying relevant context factors that influence effectiveness and implementation, leading to a mid-range theory that may inform the scaling up the intervention.

**Ethics and dissemination** In developing this protocol, we paid specific attention to cultural sensitivity, the *do no harm* principle, confidentiality and non-attribution. We received ethical approval from the local and national institutional review boards in Tanzania, Uganda, Malawi,

Benin, Sweden and Belgium. Written or verbal consent of respondents will be secured after explaining the purpose, potential benefits and potential harms of the study using an information sheet. The results will be disseminated through workshops with the hospital staff and national policymakers, and scientific publications and conferences.

**Trial registration number** PACTR202006793783148.

## INTRODUCTION

Child mortality and mortality related to pregnancy and childbirth remain major public health priorities. Globally, 5 million children die before their second birthday[1] and nearly 7.5 million children die within the first 1000 days of life. Over 2 million babies are stillborn each year[2] and for almost 300 000 women annually, pregnancy or childbirth are fatal.[3] While there is strong evidence available related to what ought to be made available to women and their new-borns during childbirth,[4] there is a lack of evidence on how best to support health providers to implement the evidence-based practices which can save lives and prevent suffering. The Action Leveraging Evidence to Reduce perinatal morTality and morbidity (ALERT) in sub-Saharan Africa (SSA) project was developed to close the gap between evidence and practice. ALERT is a 5-year multifaceted health system intervention targeting the intrapartum period. It is funded by the European Commission under the Horizon 2020 framework and led by Karolinska Institute in collaboration with seven other institutions. National universities and healthcare professional organisations in cooperation with local district health management teams and local training institutions will deliver the ALERT intervention.

The project aims to (1) overcome the knowledge–implementation gap at the level of multidisciplinary teams providing maternity care and (2) promote implementation of evidence-based interventions in hospitals with a target primary outcome of reduced in-facility perinatal (stillbirths and early neonatal) mortality. The ALERT project will develop an intervention that includes four main components (figure 1):

1. Codesign of the intervention via end-user participation, involving narratives of women, families and midwifery providers. These include nurses, nurse-midwives, midwives, auxiliary staff, physicians and obstetricians.
2. Competency-based training for midwives
3. Quality improvement (QI) cycles to help providers identify and overcome problems, supported by data from a clinical perinatal e-registry.
4. Mentoring maternity unit leaders to foster empowerment and leadership, complemented by biannual co-ordination and accountability meetings at district level.

The intervention is based on a framework that builds on many years of experience working in the respective settings and the literature on QI and learning (including, for instance, the work of Rowe et al[5] and Walker et al[6]). It was deliberately designed to integrate the above-described four components to support healthcare workers in providing safe and respectful intrapartum care. This combination is expected to leverage previous experience suggesting that in the ALERT settings, knowledge deficits remain an important bottleneck and that this may create better engagement into QI.[6 7] The QI component will

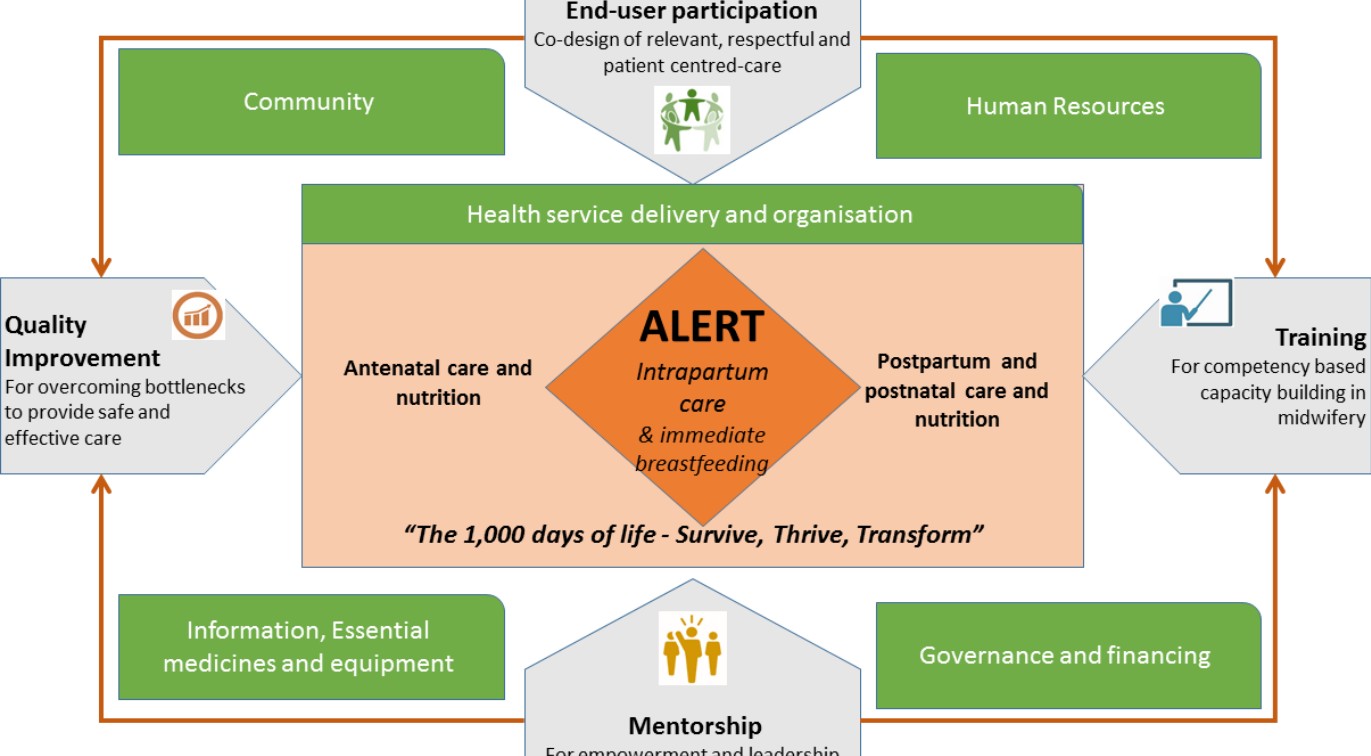

**Figure 1** The ALERT intervention. ALERT, Action Leveraging Evidence to Reduce perinatal morTality and morbidity.

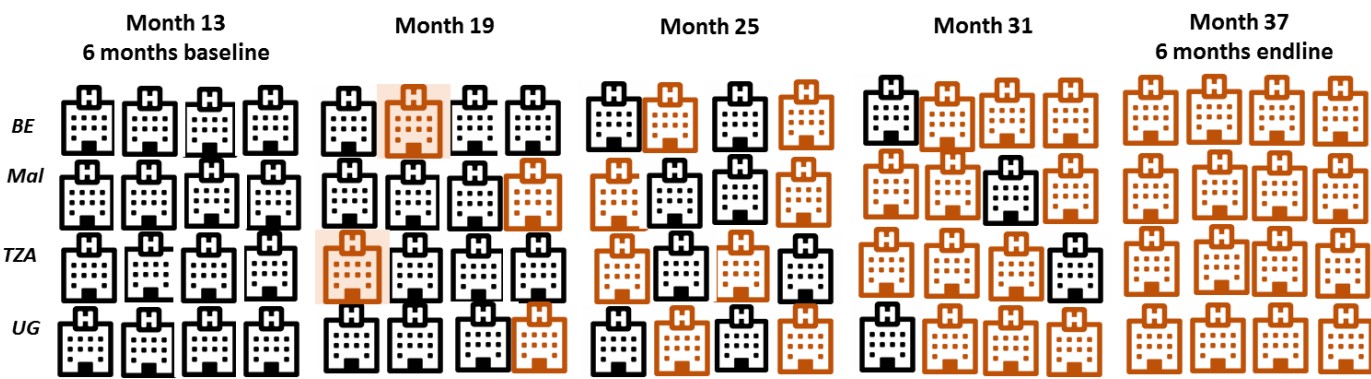

**Figure 2** Sequential integration of hospitals into the trial.

build on the work and processes put in place by existing QI teams and death review committees. Their inputs will guide the identification of the topics to be covered by the QI component, which will be aligned to the training for the midwifery care providers and use the Do-Study-Act cycle. Mentoring by valued mentors, an essential aspect of QI,[8] will be carefully designed and include a cascade approach to meet the needs of the healthcare workers. While the ALERT project is based on global evidence, the intervention will, indeed, be adapted to the context and needs of each hospital, integrating the perspectives of midwifery providers as well as women and their families. The intervention was field tested during a visit to a regional hospital in Dar-es-Salaam, Tanzania in February 2019.

The implementation started in Benin, Malawi, Tanzania and Uganda in 2020 and will run until 2024, steered by seven scientific work packages (WPs). Four district-level hospitals were selected in each country, representing a mix of public and private not-for-profit facilities. Each hospital has at least 2.500 births per year and this selection should provide sufficient power (75%–80%) to detect a 25% reduction in early perinatal mortality with 95% CIs. We refer to the overall study protocol paper for more details.[9] The intervention will be rolled out as indicated in figure 2. The hospitals were randomly selected for inclusion. More background information for each country is provided in online additional file 1.

An implementation science approach is at the heart of this study. Since the implementation of evidence-based guidelines (and arguably any QI strategy) is both determined by providers, funding and healthcare system capacity,[10] and also by its adoption and implementation by individuals and teams, it is critical to understand what works and why.[11 12] Three approaches will be employed to evaluate the ALERT intervention:

(i) A stepped-wedge design to evaluate the effect of the intervention on in-facility perinatal mortality with a primary outcome of reduced in-facility perinatal (stillbirths and early neonatal) mortality. (ii) A realist process evaluation (PE) to understand what works, for whom and under which conditions, complemented by an economic evaluation. (iii) Economic evaluation focused on scalability including costing studies in the countries.

In this paper, we present the research protocol of the realist PE, which is part of a stepped wedge trial. PEs are often carried out in implementation research, both to address the issue of the black box between an intervention and its results, and also because they are considered as useful in the design and testing of complex interventions.[13] A PE considers the actual implementation of the programme, the underlying mechanisms of change and the effect of context on outcomes.[14] To this end, PEs often incorporate an assessment of implementation (fidelity), a stakeholder analysis and a context mapping. We developed the PE on the basis of realist evaluation (RE) because of its explanatory potential: realists assume that interventions lead to effects in specific contexts, by triggering mechanisms of change for specific groups of actors. This configurational approach to causation allows for greater in-depth understanding of why and for whom intervention work (or not).[15] Over the years, RE has become recognised in health policy and systems research as a useful methodology for evaluating complex interventions.[16–18] This study contributes to filling a gap, as few studies in the field of implementation science have adopted the RE approach. We followed the Realist And MEta-narrative Evidence Syntheses: Evolving Standards (RAMESES II guidance for reporting on REs,[19] which we have adapted to fit the format for a protocol paper (see online additional file 2).

## METHODS AND ANALYSIS
### The overall methodological approach
In the quest to open the 'black-box' of complex programmes, Pawson and Tilley developed RE, a theory-driven research approach that seeks to understand the links between context, outcomes and mechanisms.[20] While traditional evaluation questions often focus on effectiveness or on process, RE seeks to answer a string of questions: *what works, for whom, in what respect, to what extent, in what context and how?* From the perspective of RE, programmes work (or not) because actors make decisions in response to the opportunities or resources that the programme provides. According to Pawson and Tilley, programmes or interventions can be viewed as theories or assumptions held by people (including

designers, implementers, beneficiaries and policy-makers) who are embedded in the open social systems in which they operate. Therefore, it is not the programme, but the actors who are engaged in it who bring about the results. To understand how that happens, RE explores the context in which the programme operates and the mechanisms that are underlying the observed change (or the lack thereof). Mechanisms refer to psychological, social, cultural or organisational factors that explain why actors respond to a programme the way they do. This response, however, does not happen in isolation; the context in which the programme takes place influences the way actors respond to it: mechanisms (and thus the programmes that trigger them) only work if the context conditions are right. Summarily, not every programme works identically in different settings. Realist evaluators use the term *programme theory (PT)* to describe the explanation of how a programme has contributed to observed results. The PT is both the end product of a RE, and it is also a hypothesis that serves as the starting point of any RE. The initial PT shapes the study design, data collection, analysis and synthesis.

We structured the PE of ALERT according to the steps of the realist research cycle.[21]

► In consideration of the topic being researched, realists build the *initial PT* on existing evidence (eg, literature reviews, document reviews), interviews with key actors and/or exploratory research.

► The *study design* is chosen as a function of the PT that needs to be assessed.

► *Data collection* tools and methods are selected based on the components of the initial PT to be tested: all data required to test the initial PT should be collected. Often mixed methods (qualitative and quantitative) of data collection are used.

► *Data analysis*: the Intervention-Context-Actor-Mechanism-Outcome (ICAMO) configuration heuristic is often used in the analysis of data.[15] Qualitative data analysis is usually based on a thematic coding approach using the core elements of the initial PT. The quantitative data are often analysed with descriptive statistics and integrated in the analysis using a retroductive approach.

► Each study ends with a *synthesis* based on a comparison of the findings of the study with the initial PT, which is adapted as needed, leading to a refined PT.

## Objectives
The main objective of the ALERT PE is to identify and test the causal pathway through which the intervention achieves its impact. Summarily, it aims to answer the question *Which components of the ALERT intervention work, how, for whom and in which contexts?*

There are four specific objectives:

(i) To collaboratively elicit the initial PT of the ALERT intervention. (ii) To test the initial PT through a multiple embedded case study. (iii) To refine the PT of the ALERT intervention using cross-case analysis. (iv) To summarise

and disseminate findings in the form of recommendations to inform implementation and practice (of quality intrapartum care) at hospital and country levels.

In the following sections, each of the four core objectives and the corresponding data collection and analysis methods are presented.

### Objective 1: defining the initial programme theory
We will formulate the initial ALERT PT based on its theory of change (TOC) (see figure 3), a review of the existing evidence on adoption and diffusion of innovations, and the perspectives of stakeholders involved in the intervention.

ALERT's TOC is informed by the integrated Promoting Action on Research Implementation in Health Services (i-PARIHS) framework.[22] It identifies the *what* (ie, an intervention that fits with existing practice, is user-friendly, has a comparative advantage and can be tested); the *who* or *recipients* (ie, end users: midwifery providers, mothers and their families, who are motivated and have sufficient skills and resource support by collaboration and teamwork); the *where* or *context* (SSA hospitals with formal and informal leadership support, effective evaluation and feedback processes and a conducive learning environment) and the *how* (ie, a problem-focused multi-faceted approach to ensure participation and involvement, ownership and control and empowerment within and between groups). The TOC implies that end-user participation ensures that the needs of women, families and midwifery providers (ie, the end users) are identified and addressed in the form of training and QI. Also, better trained and supported midwifery providers will provide better care, including improved foetal monitoring, improved emergency care and client-centred support during labour. Furthermore, better skilled and empowered midwifery providers will also be better able to quickly make decisions. Finally, improved hospital processes leading to timely access to caesarean section or operative vaginal delivery will reduce hypoxic–ischaemic events and thus reduce stillbirths and neonatal deaths.

In order to move the TOC to the level of an initial PT, we would need to identify the context elements and mechanisms that would explain how and why the ALERT intervention would achieve the expected results. We plan to carry out the following steps:

► We will summarise existing evidence to identify any theories that may explain how and in which conditions the ALERT intervention will be effective. More concretely, we will focus on the adoption, diffusion and maintenance of the intervention (components) by healthcare providers and hospital managers and the role of multistakeholder co-production in these processes.

► We will draw on health facility assessment data, stakeholder mapping data and the national context assessment data collected by other WPs in the intervention, to describe the national and facility context, and the views and perspectives of national stakeholders.

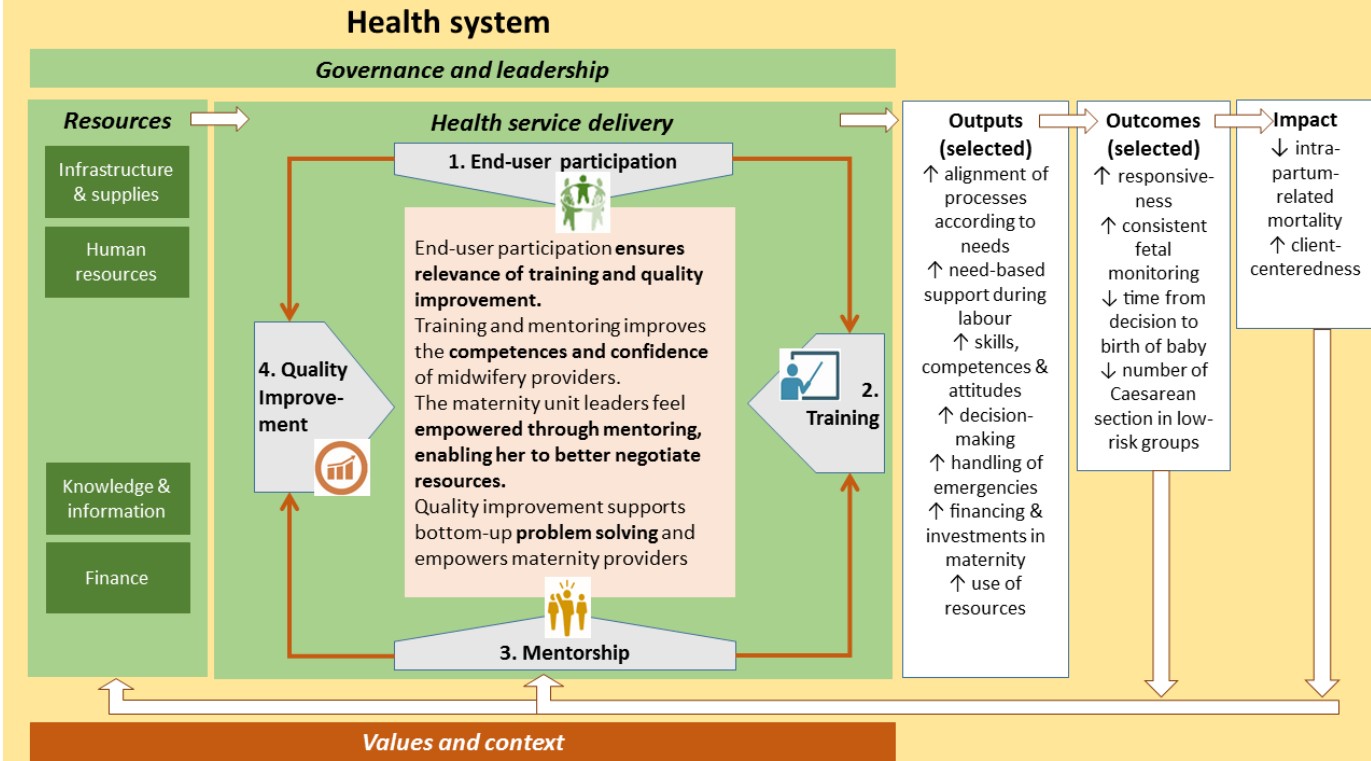

**Figure 3** The ALERT theory of change. ALERT, Action Leveraging Evidence to Reduce perinatal morTality and morbidity.

► We will organise participatory stakeholder workshops to elicit the assumptions of the ALERT consortium members.
► We will use the opportunity of online and face-to-face monthly coordination meetings, consortia meetings, discussions and feedback sessions to present and discuss the evolving initial PT to our colleague-researchers and stakeholders from each country.

The end product of these is the initial PT that will guide the further steps of the PE.

### Objective 2: a multiple embedded case study to test the initial PT

To test the initial PT, we will use a case study design. Investigating the intervention's implementation through case studies in hospitals in four countries will allow us to monitor the actual implementation of the intervention and how it progressively evolves (in terms of actual implementation, context conditions, response of actors and potential mechanisms) across the study sites.

### The case study design

We will carry out a multiple embedded case study.[23] We define 'the case' as the implementation of the ALERT intervention. We define 'study site' as a hospital implementing the ALERT intervention. In RE, study sites are chosen purposively in order to allow testing the initial PT (ie, theoretical sampling). In practice, in each country, we will select two of the four study sites, whereby we aim to choose contrasting sites that are likely to 'manifest' the IPT differently due to contextual variation.

In each country, the first site selected for the realist PE is the hospital where the ALERT intervention will start

at step 1 (Month 19 (M19)–M24) of the stepped wedge intervention design (figure 2). In these four hospitals, we will conduct a detailed 'thick' evaluation (see details on data collection below). The second hospitals will be chosen among the stage 3 hospitals (M31–M36 of implementation), and a 'light' evaluation will be conducted. The phased recruitment of hospitals into the realist PE will enable the assessment of how length of exposure to the intervention and changes over time influence outcomes. Selecting two hospitals per country will allows us to conduct in-country cross-case comparisons and to identify how mechanisms play out differently at country and facility levels.

### Data collection

We will use the following data collection methods and tools:
► In each site, we will carry out in-depth interviews with health stakeholders (including midwives, maternity unit managers, hospital managers and the district director of health), focusing on their view on the reasons the ALERT intervention works (or not).
► This will be complemented by opportunistic observations and review of relevant hospital documents.
► We will draw on data collected by WP3 (training module development), WP4 (health facility assessment), WP5 (perinatal data) and WP 6 (evaluation and fidelity monitoring) to describe the planned intervention (end product of WP3), the actual implementation of the intervention (WP6), as well as the context and the actors (WP4) and the processes triggered by

**Table 1** Summary of ALERT realist process evaluation data collection

| Specific objective | Data collection methods | Product/finality |
|---|---|---|
| Defining the initial programme theory | 1. Scoping review<br>2. ALERT consortium stakeholder workshop<br>3. Country level stakeholder workshops (x4)<br>Data used from other work packages<br>1. Health facility assessments and Bottleneck analysis<br>2. Implementation process description | The initial programme theory of the ALERT intervention |
| Multiple embedded case study to test the IPT | 1. Context mapping (hospital)<br>2. Context mapping (national/health system)<br>3. In-depth interviews with women and their families<br>4. In-depth interviews with midwives, maternity unit managers, hospital managers, district directors of health<br>Data used from other work packages<br>1. Review data collection by WP 2 and 3 from narratives and ethnographic observations of women and health providers | Site-specific ICAMO configurations and potentially programme theories |
| Cross-case analysis to refine the programme theory of ALERT | This step involves first the comparison of the country-specific programme theories in order to identify patterns of similarities and differences.<br>In a second step, the findings are brought to a next level of abstraction, aimed at capturing the essential elements that explain the results of each site in a comprehensive manner | 1. Country-specific programme theories. Overarching programme theory |

ALERT, Action Leveraging Evidence to Reduce perinatal morTality and morbidity; ICAMO, Intervention-Context-Actor-Mechanism-Outcome; WP, work packages.

the intervention (WP6). For more details about these WPs, we refer to the overall protocol paper.[9]

► We will carry out in-depth interviews with mothers and their families to document their views on the design and implementation of the ALERT intervention.

► We will draw on the data from patient exit interviews carried out by another ALERT WP to obtain additional information on the views of patients on the care they received and the (potential) influence of the ALERT intervention.

Intervals between data collection offer time to analyse data and refine the IPT based on emerging patterns which will be presented at the subsequent consortium meetings. Table 1 summarises the data collection plan.

### Data collection tools

For each data collection method described above, specific tools will be used: a facilitator guide for the participatory stakeholder workshops for ALERT consortium members; in-depth interview guides for hospital managers/in-charges, midwives and district directors of health and in-depth interview guides for mothers and families (the latter interviews will be carried out within another WP). Local research assistants will collect data in each country. To bridge gaps in technical capacity, especially with respect to realist interviewing techniques,[24] we will conduct face-to-face training, complemented by remote training and support.

### Data management

Audio files (and handwritten notes) will be translated, transcribed verbatim and managed for analysis using NVIVO qualitative data analysis software. Translation will aim to retain technical and cultural meaning in order to minimise interpretation gaps resulting from cross-cultural differences.[25] A random selection of audio files and their corresponding transcripts will be reviewed and checked for quality and accuracy by a member of the realist PE team. Audio files and interview transcripts will be tagged alpha-numerically based on date, district identifier and category of stakeholder. Individuals will not be identified by their real names in field notes. After each data collection session, all information (raw and transcribed data) will be stored on password-protected devices (external drives, computers) and will only be accessed by evaluation team members.

### Data analysis

As mentioned above, RE is method-neutral: discipline-specific and appropriate techniques are adopted during data analysis. In general, the analysis will be guided by the ICAMO configuration. Analysis of the qualitative data will be guided by the primary elements and themes identified in the initial ALERT PT. Analysis of transcripts will begin immediately following transcription, allowing for early identification of ICAMO configurations and patterns. In line with realist principles, a thematic coding approach will be used that is based on the core elements of the initial PT. The framework analysis method[25] suits the realist approach as it allows for the inclusion of both a priori and emergent concepts.

► The data will be categorised in a first round of analysis using the ICAMO configuration.

- ► New interpretations will emerge in subsequent rounds of coding, leading to a refined analysis.
- ► Next, a retroduction approach will be adopted, whereby the observed outcomes are explained by looking into the mechanisms, actual intervention modalities, actors and context elements. This results in descriptions of the actual intervention, its effects (both intended and unintended, positive and negative), the context elements and the underlying mechanisms (ICAMO configurations), which offer the most plausible explanations of the observed outcomes.

During this analytical process, we will also draw on data and findings from other WPs and integrate participatory reflection and feedback from the ALERT consortium annual meetings. Data triangulation and verification will be ensured by comparing responses from different respondents and from other data sources in order to identify similar and divergent views. This will contribute to the internal validity of the analysis.

### Synthesis

The site-specific ICAMO configuration(s) will be compared with the initial PT, and in a process of abstraction, the initial PT will be modified when and where needed.

### Objective 3: cross-case analysis to refine the programme theory of ALERT

The above described 'in-case' analysis will be followed by cross-case analysis within and between the countries. Comparing between the study countries should enable identifying relevant context factors at country level and assessing their influence on intervention implementation and scale up. In practice, we will develop tables presenting the different ICAMO configurations by country. We will look for patterns of similarity and difference and see how these challenge, refute or confirm the initial PT. Prior to the finalisation of the refined PT, the results of the PE will be discussed in a final round of participatory workshops with ALERT partners and in-country stakeholders directly involved in the project. This will provide an opportunity to contextualise, reflect on and improve the PT.

In summary, the final round of PT development will occur iteratively during the cross-case analysis, leading to a final PT that may reach the level of a mid-range theory, which indicates what it is about the ALERT intervention that works for whom and in which circumstances. The results may both inform the formulation of recommendations for scaling up the intervention in the study countries, and may also allow tailoring of similar interventions to different contexts.

### Objective 4: dissemination

To summarise and disseminate findings in the form of recommendations at study site and country level, we will leverage the annual consortium project meetings. This also serves as an opportunity for cross-pollination of experiences and learning between intervention countries, and

for presenting and discussing the research findings within the broader findings of the project. Feedback meetings will adopt a participatory open discussion approach with the aim to discuss potential and critical action points for improving implementation and scaling up to other locations, as well as the possible implications (positive and negative) of such efforts.

In addition to the final country reports on the RE from each country, findings will be further disseminated through at least two open-access publications and two international conferences. We expect that the doctoral student working on this PE will provide additional avenues for dissemination at country and international level (conferences and peer-reviewed publications).

### Patient and public involvement

In this component of the ALERT project, which in essence is an evaluation of a QI intervention, patients nor members of the public were involved in the design and they will not be involved in the actual evaluation and analysis. However, during the formative phase of the project, the ALERT intervention will be codesigned with women and families living in the study sites. Their narratives regarding pregnancy, birth and their health seeking behaviour in general will be documented and analysed in order to inform the intervention design. They will be included in feedback meetings at the end of the project.

### Ethics and dissemination

The research team recognises the ethical implications of any form of research. The realist PE will be carried out in line with the principles of the Declaration of Helsinki as amended in 2013, and any further versions.[26]

### General principles

The study team will place particular emphasis on:

### Cultural sensitivity

Data collection modalities will be aligned to the socio-cultural norms and preferences of people in their daily life and within organisational settings (eg, ensuring interviews with health workers does not disrupt health service provision and other duties). All personnel on the study team will be bound to the code of conduct and ethical restrictions of the research.

### Do no harm

Respondents will be well informed of their rights to withdraw from the study at any point in time and all efforts will be made to ensure that no discomfort or stress is experienced by respondents as a result of the research. This includes allowing respondents to choose their preferred location and time for interviews with care taken to ensure privacy to the greatest degree possible. Informed consent, privacy and confidentiality, as well as respect for participants' time will be prioritised.

## Confidentiality and non-attribution

Only ALERT team members will have access to raw data. All documents, completed questionnaires and audio-recordings will be treated with the strictest care for confidentiality. Reports will be compiled with the intention to protect the identity of respondents while representing their views and opinions as accurately and fairly as possible.

## Vulnerability of the target populations

The views and experiences of respondents for the realist PE are primarily collected through participatory sessions and in-depth interviews (ie, qualitative methods). We consider that the issues to be discussed in interviews are of a non-sensitive nature and we do not foresee any harmful risks or adverse outcomes (principle of non-maleficence). Due to the nature of health workers' jobs, interviews will be conducted at a place and time of their convenience without hindering their service delivery tasks. Respondents will, however, be making a sacrifice of their time in order to participate in the interviews. They will be compensated in line with the country standards for transportation reimbursement and provision of refreshments (water, non-alcoholic beverages and snacks).

## Informed consent

Written or verbal consent of respondents will be secured after explaining the purpose, potential benefits and potential harms of the study using an information sheet. Copies of informed consent will be available in the local language of the participants (French, Kiswahili or Chichewa) and where applicable, English. Data collectors will explain the informed consent form and respondents will be encouraged to ask for clarification or ask any questions. Respondents able to read and write will be required to sign the consent form themselves. In case of illiterate study participants, the data collectors will read the form carefully to them and they will countersign the form using their thumb impression. Where participants prefer to provide verbal consent in place of written consent, the consent will be audio-recorded. In this case, the participant or the researcher will read the paragraph granting consent aloud while being audio-recorded. Interviews conducted under verbal consent can only proceed if there is at least an audio-recording of the consent. The respondent can still decline audio-recording for the full interview. Where the respondent does not give consent for audio-recording, only handwritten notes will be taken during the interview.

## Process of withdrawal from the study

The consent procedures make it clear to research participants that they have the right to withdraw from the study at any point, and to refuse to answer questions without any negative consequences.

Possibilities for study withdrawal would include the following:

- ► Where respondents have identified that they would be at risk from participating in the study
- ► Where a respondent self-reports that there has been a threat, verbal abuse or attempts to restrict their access to care or other services due to their participation in the data collection.
- ► When the research team identifies that there has been an incident, where recriminations have been levelled against respondents due to their participation in the study.
- ► Where a potential respondent declines providing any type of documented consent (written or verbal).

We received ethical approval from the local and national institutional review boards as follows: Karolinska Institutet, Sweden (Etikprövningsmyndigheten Dnr 2020-01587); Uganda National Council for Science and Technology (UNCST) (HS1324ES); Muhimbili University of Health And Allied Sciences (MUHAS) Research and Ethics Committee, Tanzania (MUHAS-REC-04-2020-118) and The Aga Khan University Ethical Review Committee, Tanzania (AKU/2019/044/fb); College of Medicine Research and Ethics Committee (COMREC), Malawi (COMREC P.04/20/3038); Comité National d'Ethique pour la Recherche en Santé, Cotonou, Bénin (83/MS/DC/SGM/CNERS/ST); The Institutional Review Board at the Institute of Tropical Medicine Antwerp and The Ethics Committee at the University Hospital Antwerp, Belgium (ITG 1375/20. B3002020000116).

## Dissemination

The results of this study will be published following the European Union's open-access policy. As mentioned above, we will use several avenues. First, we will organise workshops at hospital-level, district-level and national level in the four study countries. Second, we will draft and submit scientific publications to open access journals and submit our work for presentation at conferences. Third, all publications and products will be published in a website (alert.ki.SE).

## DISCUSSION

In implementation science, finding heterogeneity in results across different settings and types of organisations and by different actors is common. This variation may often be explained by the dynamic interplay between the intervention, its context, the implementation process and the agency of involved actors. When these interactions are not accounted for in an evaluation, the result is a 'black box' of knowledge that provides little or no explanatory insight on causal factors that explain how interventions lead to outcomes. In response to the evaluation approaches dominant in the 1970s and which focused almost exclusively on effectiveness, theory-driven inquiry methods were developed.[27] Theory-based evaluation,[28] theories of change[29] and RE[20] all present methods to open the black box between intervention and outcome.

The 2015 guidance on the conduct of PE developed by the UK Medical Research Council (MRC) calls for paying more attention to the intervention theory.[13] The updated guidance published in 2021 reinforced this and emphasised also the role of context.[30] This resonates with a similar shift in the field of implementation research, which is increasingly using conceptual models, frameworks and theories to design interventions but also to guide the design of evaluations, interpret findings and enhance translation of research into policy and practice.[31] Kislov *et al* call for using mid-range theories of implementation, referring to Merton's definition of middle-range theories as theories that are situated between minor working hypotheses (nowadays often referred to as programme theories) and the all-inclusive unified theory (or grand theories).[32] Yet, the use of theory in PEs remains limited. In their recent systematic review of PEs embedded in trials, McIntyre and colleagues found that in about 66% of the reviewed papers, the authors stated to use theory. Yet only in 26% of the reviewed studies, theoretical propositions are actually applied, and in only 7% of the studies did the authors test theory.[33]

Previous studies have shown that realist principles can be adapted and applied to all phases of the PE cycle.[34] A realist PE may offer two major advantages. First, it offers the possibility to systematically test the intervention theory by grounding it in the existing body of knowledge and testing it empirically, augmenting its explanatory power in the process.[35] Second, it allows to build up theories by accumulating empirical insights that push the PT to the level of a mid-range theory. Such mid-range theory has explanatory power extended to other settings, thereby offering the possibility to inform decisions on introducing or scaling up similar interventions.[18] Within the ALERT project, carrying out a realist PE across eight hospitals in four countries will allow us both to better understand the differential effects of the ALERT intervention in different hospitals in different countries, and also to contribute to the methodological developments within the field of PEs.

As with any study, there are some limitations to this study. Social desirability bias can occur when respondents answer to questions based on what they believe the interviewer wants to hear, or when specific responses are assumed to be more socially acceptable. Attempts to minimise this bias will be made by asking to view documentation where appropriate and triangulating data from multiple sources across WPs. The challenges of cross-cultural differences, power imbalances and western-dominated theories in realist research are additional recognised risks.[36] Approaches to mitigate this include the use of local researchers guided by a trained doctoral student with oversight provided by a principal investigator who is well-versed in realist methodology; contextualisation and validation of data through participatory workshops and iterative meetings; and continuous reflection by evaluation team members on their positioning within the research study and potential sources of bias.

**Author affiliations**
[1]International Program Evaluation Unit, Centre for Global Child Health, Hospital for Sick Children, Toronto, Ontario, Canada
[2]Division of Social & Behavioural Health Sciences, University of Toronto Dalla Lana School of Public Health, Toronto, Ontario, Canada
[3]Department of Public Health, Institute of Tropical Medicine, Antwerpen, Belgium
[4]Centre de Recherche en Reproduction Humaine et en Démographie, Cotonou, Benin
[5]Department of Public Global Health, Karolinska Institute, Stockholm, Sweden
[6]University of Malawi College of Medicine, Blantyre, Malawi
[7]Muhimbili University of Health and Allied Sciences, Dar es Salaam, Tanzania
[8]Department of Health Policy Planning and Management, Makerere University, Kampala, Uganda
[9]Global Public Health, Karolinska Institute, Stockholm, Sweden

**Acknowledgements** We would like to acknowledge the support of the Action Leveraging Evidence to Reduce perinatal morTality and morbidity study team, which is a consortium of researchers and implementers of eight institutions across Europe, Benin, Malawi, Tanzania and Uganda.

**Collaborators** This group developed the Action Leveraging Evidence to Reduce perinatal morTality and morbidity WP work packages (ALERT) project and is responsible for the implementation and the evaluation of the multifaceted intervention. The composition of the group is as follows.The ALERT Study Team: Ahossi Angèle Florence Laure, Andrea B Pembe, Ann-Beth Nygaard Moller, Antoinette Sognonvi, Armelle Vigan, Banougnin Bolade Hamed, Beatrice Mwilike, Kéfilath Bello, Bianca Kandeya, Christelle Boyi Metogni, Bruno Marchal, Claudia Hanson, Dickson Mkoka, Effie Chipeta, Elizabeth Ombeva Ayebare, Fadhlun M Alwy Al-Beity, Gertrude Namazzi, Gisele Houngbo, Gottfried Agballa, Hashim Hounkpatin, Helga Naburi, Helle Mölsted Alvesson, Hussein L Kidanto, Jean-Paul Dossou, Joanne Welsh, Joseph Akuze, Josephine Babirye, Kristi Sidney Annerstedt, Lenka Benova, Lilian Mselle, Mechthild Gross, Muzdalifat Abeid, Nicola Orsini, Peter Waiswa, Philip Wanduru, Razak Mussa, Regine Unkels, Rian Snijders, Samuel Meja, Schadrac Agbla, Therese Delvaux, Tumbwene Mwansisya, Virginia Castellano Pleguezuelo, William Stones, Wim Van Damme, Yesaya Z Nyirenda and Zamoyoni Julius.

**Contributors** I-OOA and BM designed the study, wrote the study protocol, designed the data collection guides and the analytical strategy, drafted the initial manuscript and drafted the final version. VCP, LB, CH and J-PD contributed to the study design and the design of the data collection guides. I-OOA, BM, VCP, LB, CH, J-PD, CH, CBM, SM, DAM, GN and KS contributed to the initial draft of the manuscript and to the final version. I-OOA, BM, VCP and LB contributed to the revision.

**Funding** This study is part of the ALERT project which is funded by the European Commission's Horizon 2020 (No 847824) under a call for Implementation research for maternal and child health.

**Disclaimer** The funder had no role in the design of the study and collection, analysis and interpretation of data and in writing the manuscript. The contents of this article are solely the responsibility of the authors and do not reflect the views of the European Union.

**Competing interests** None declared.

**Patient and public involvement** Patients and/or the public were not involved in the design, or conduct, or reporting or dissemination plans of this research.

**Patient consent for publication** Not required.

**Provenance and peer review** Not commissioned; externally peer reviewed.

**ORCID iDs**
Ibukun-Oluwa Omolade Abejirinde http://orcid.org/0000-0003-0139-0541
Claudia Hanson http://orcid.org/0000-0001-8066-7873
Bruno Marchal http://orcid.org/0000-0001-7185-022X

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
