## [Reviewer comments · BMJ Open]

ARTICLE DETAILS

TITLE (PROVISIONAL)	Strengthening capacity in hospitals to reduce perinatal morbidity and mortality through a co-designed intervention package: protocol for a realist evaluation as part of a stepped-wedge trial of the Action Leveraging Evidence to Reduce perinatal morTality and morbidity in sub-Saharan Africa (ALERT) project
AUTHORS	Abejirinde, Ibukun-Oluwa; Castellano Pleguezuelo, Virginia; Benova, Lenka; Dossou, Jean-Paul; Hanson, Claudia; Metogni, Christelle Boyi; Meja, Samuel; Mkoka, D; Namazzi, Gertrude; Sidney, Kristi; Marchal, Bruno

VERSION 1 – REVIEW

REVIEWER	Semrau, Katherine Ariadne Labs at Brigham and Women's Hospital and the Harvard T.H. Chan School of Public Health
REVIEW RETURNED	27-Oct-2021

GENERAL COMMENTS	Thank you for the opportunity to review the manuscript, "Action Leveraging Evidence to Reduce perinatal morTality and morbidity in sub-Saharan Africa (ALERT): a protocol of a realist evaluation." The paper is well-written and detailed. I found it not only a protocol for the ALERT study, but a methods paper providing insights to realist evaluation approaches and strategies. A few minor comments for consideration by the authors: 1. The abstract focuses primarily on the methods being used; however, the paper presents details around the topical areas being addressed in the intervention (particularly, competency-based midwife training, community engagement for study design, mentoring, and QI cycles). These interventions are not currently shared in the Abstract, but readers could relate more concretely to the paper if highlighted in the abstract.2. In the introduction, can the authors clarify if the intervention is designed at the hospital level or are the two hospitals in the same country receiving the same intervention? Further, from a clinical perspective, are ob/gyns and nurses included in the intervention or only midwives?3. Given the study initiation around Feb 2019 (pilot study) and work over the past 24 months, how has the COVID-19 pandemic altered their protocol? This could be useful for others who are developing protocols now.4. Please consider adding a statement on intent for data availability; clearly this study will produce a large amount of
--

	qualitative and quantitative data. This could be included as part of the ethics and dissemination section.
--	--

REVIEWER	Pearson, Mark University of Hull, Hull York Medical School
REVIEW RETURNED	19-Feb-2022

GENERAL COMMENTS	This protocol clearly positions the study to address unknowns about how to implement evidence-based practice in perinatal care. The rationale for conducting a realist process evaluation is clearly stated. I note the completion (appropriately adapted for a protocol) of the RAMESES II reporting standards table. For me, the title is somewhat misleading as it implies that the whole study is a realist process evaluation – more accurately, it’s a realist process evaluation as part of a stepped-wedge trial (e.g. see the title of Ref 35 cited by the protocol). The introduction mentions ‘Quality Improvement’ without any description of what QI is. This is important to address as the text on p.6 repeatedly refers to the significant role of ‘QI’ without explaining what ‘QI’ involves and therefore what role it is playing in the intervention. Better explanation here would also make the condensed presentation of the intervention in Figure 1 more comprehensible, and would ease any concerns raised later on p.17 where it is stated that the RPE is “in essence an evaluation of a quality improvement intervention”. Methods: Selection of study sites – this comes across as a little ‘off-hand’. It is of course reasonable that the process evaluation purposively samples from the sites at which the intervention is being delivered, but the process through which study sites for the stepped wedge trial were selected should also be stated. Data collection – it is stated that “data collected by other work packages to describe intervention, context, actors...” will be drawn upon. More detail about precisely what data this will be should be provided – these aspects do not appear to be ‘nice to have additions’, rather they are central to the realist goal of providing explanatory insight. The (re-)justification for a theory-driven approach in Paragraphs 2 and 3 of the Discussion section seems somewhat mis-placed – it would be more appropriate (in shorter form) as part of the Introduction or Methods. The authors may also wish to take the opportunity to situate the study in relation to the recently-revised MRC Complex Interventions Guidance: Skivington, K., L. Matthews, S. A. Simpson, P. Craig, J. Baird, J. M. Blazeby, K. A. Boyd, N. Craig, D. P. French, E. McIntosh, M. Petticrew, J. Rycroft-Malone, M. White and L. Moore (2021). "A new framework for developing and evaluating complex interventions: update of Medical Research Council guidance." BMJ 374: n2061 Both the Discussion (end of manuscript) and ‘Strengths and limitations’ summary (section at the start of the manuscript) could articulate the distinct contribution of the study to improving perinatal care in sub-Saharan Africa much more clearly.
--

	Keywords are all methodological. Surely keywords should include area of health care, geographical region, and so on.
--	--

VERSION 1 – AUTHOR RESPONSE

Reviewer: 1

Dr. Katherine Semrau, Ariadne Labs at Brigham and Women's Hospital and the Harvard T.H. Chan School of Public Health, Harvard Medical School

Comments to the Author:

Thank you for the opportunity to review the manuscript, "Action Leveraging Evidence to Reduce perinatal morTality and morbidity in sub-Saharan Africa (ALERT): a protocol of a realist evaluation." The paper is well-written and detailed. I found it not only a protocol for the ALERT study, but a methods paper providing insights to realist evaluation approaches and strategies. A few minor comments for consideration by the authors:

Thank you for these positive remarks.

1. The abstract focuses primarily on the methods being used; however, the paper presents details around the topical areas being addressed in the intervention (particularly, competency-based midwife training, community engagement for study design, mentoring, and QI cycles). These interventions are not currently shared in the Abstract, but readers could relate more concretely to the paper if highlighted in the abstract.

We adapted the abstract as follows:

“Despite a strong evidence base for developing interventions to reduce child mortality and morbidity related to pregnancy and delivery, major knowledge-implementation gaps remain. The Action Leveraging Evidence to Reduce perinatal morTality and morbidity in sub-Saharan Africa (ALERT) project aims to overcome these gaps through strengthening the capacity of multidisciplinary teams that provide maternity care. *The intervention includes competency-based midwife training, community engagement for study design, mentoring, and Quality Improvement cycles.* The realist process evaluation of ALERT aims at identifying and testing the causal pathway through which the intervention achieves its impact.”

2. In the introduction, can the authors clarify if the intervention is designed at the hospital level or are the two hospitals in the same country receiving the same intervention? Further, from a clinical perspective, are ob/gyns and nurses included in the intervention or only midwives?

To the description of the intervention, we added: “These include nurses, nurse-midwives, midwives, auxiliary staff, physicians and obstetricians “

We also added the following sentence to the *Introduction* : “While the ALERT project is based on global evidence, the intervention will be adapted to the context and needs of each hospital.”

3. Given the study initiation around Feb 2019 (pilot study) and work over the past 24 months, how has the COVID-19 pandemic altered their protocol? This could be useful for others who are developing protocols now.

There were indeed a few consequences, but we are not sure this merits being put in the manuscript, as the changes were rather ‘generic’. When the COVID-19 pandemic was declared, the ALERT study protocol was being finalised for submission to the ethics committees. The study was consequently affected in several ways. First, we added questions on the impact of COVID-19 on the involved hospitals were in the health facility assessment tool (WP4). Second, the timeline for data collection for co-design (WP2), training and mentoring (WP3), quality improvement (WP4), and e-registry development (WP5) was delayed. Overall, the start of the trial was delayed with 6 months. Third, the international meetings scheduled for the whole research consortium members and their visits to the participating hospitals have been negatively affected. Virtual meetings with the international consortium members were set up instead.

4. Please consider adding a statement on intent for data availability; clearly this study will produce a large amount of qualitative and quantitative data. This could be included as part of the ethics and dissemination section.

The editor requested us to delete any statement on data availability and sharing.

Reviewer: 2

Dr. Mark Pearson, University of Hull

Comments to the Author:

This protocol clearly positions the study to address unknowns about how to implement evidence-based practice in perinatal care. The rationale for conducting a realist process evaluation is clearly stated. I note the completion (appropriately adapted for a protocol) of the RAMESES II reporting standards table.

For me, the title is somewhat misleading as it implies that the whole study is a realist process evaluation – more accurately, it's a realist process evaluation as part of a stepped-wedge trial (e.g. see the title of Ref 35 cited by the protocol).

We agree and revised the title as follows (even if it has become quite long): “Strengthening capacity in hospitals to reduce perinatal morbidity and mortality through a co-designed intervention package: the protocol of a realist evaluation as part of a stepped-wedge trial of the Action Leveraging Evidence to Reduce perinatal morTality and morbidity in sub-Saharan Africa (ALERT) project”.

The introduction mentions ‘Quality Improvement’ without any description of what QI is. This is important to address as the text on p.6 repeatedly refers to the significant role of ‘QI’ without explaining what ‘QI’ involves and therefore what role it is playing in the intervention. Better explanation here would also make the condensed presentation of the intervention in Figure 1 more comprehensible, and would ease any concerns raised later on p.17 where it is stated that the RPIs “in essence an evaluation of a quality improvement intervention”.

We agree and revised the description of the intervention as follows:

“The intervention is based on a framework that builds on many years of experience working in the respective settings and the literature on quality improvement and learning (including, for instance, the work of Rowe *et al.*⁵ and Walker *et al.*⁶ It was deliberately designed to integrate the above-described four components to support health care workers in providing safe and respectful intrapartum care. This combination is expected to leverage previous experience suggesting that in the ALERT settings, knowledge deficits remain an important bottleneck and that this may create better engagement into QI.^{6,7} The QI component will build on the work and processes put in place by existing quality improvement teams and death review committees. Their inputs will guide the identification of the topics to be covered by the QI component, which will be aligned to the training for the midwifery care providers and use the Do-Study-Act (PDSA) cycle. Mentoring by valued mentors, an essential aspect of QI⁸, will be carefully designed and include a cascade approach to meet the needs of the health care workers. While the ALERT project is based on global evidence, the intervention will, indeed, be adapted to the context and needs of each hospital, integrating the perspectives of midwifery providers as well as women and their families. The intervention was field tested during a visit to a regional hospital in Dar-es-Salaam, Tanzania in February 2019.”

Methods:

Selection of study sites – this comes across as a little ‘off-hand’. It is of course reasonable that the process evaluation purposively samples from the sites at which the intervention is being delivered, but the process through which study sites for the stepped wedge trial were selected should also be stated.

We added a more detailed description of the selection of the study hospitals in the Introduction section, based on the published overall trial protocol: “Four district-level hospitals were selected in each country, representing a mix of public and private not-for-profit facilities. Each hospital has at least 2,500 births per year and this selection should provide sufficient power (75–80%) to detect a

25% reduction in early perinatal mortality with 95% confidence intervals. We refer to the overall study protocol paper for more details.⁹

Data collection – it is stated that “data collected by other work packages to describe intervention, context, actors...” will be drawn upon. More detail about precisely what data this will be should be provided – these aspects do not appear to be ‘nice to have additions’, rather they are central to the realist goal of providing explanatory insight.

We added more details “We will draw upon data collected by WP3 (training module development), WP4 (health facility assessment), WP5 (perinatal data) and WP 6 (evaluation and fidelity monitoring) to describe the planned intervention (end product of WP3), the actual implementation of the intervention (WP6), as well as the context and the actors (WP4) and the processes triggered by the intervention (WP6). For more details about these work packages, we refer to the overall protocol paper.”

The (re-)justification for a theory-driven approach in Paragraphs 2 and 3 of the Discussion section seems somewhat mis-placed – it would be more appropriate (in shorter form) as part of the Introduction or Methods. The authors may also wish to take the opportunity to situate the study in relation to the recently-revised MRC Complex Interventions Guidance: Skivington, K., L. Matthews, S. A. Simpson, P. Craig, J. Baird, J. M. Blazeby, K. A. Boyd, N. Craig, D. P. French, E. McIntosh, M. Petticrew, J. Rycroft-Malone, M. White and L. Moore (2021). "A new framework for developing and evaluating complex interventions: update of Medical Research Council guidance." *BMJ* 374: n2061 We agree and added the following: “The updated guidance published in 2021 reinforced this and emphasised also the role of context.³⁰”

Both the Discussion (end of manuscript) and ‘Strengths and limitations’ summary (section at the start of the manuscript) could articulate the distinct contribution of the study to improving perinatal care in sub-Saharan Africa much more clearly.

We adapted the Strengths and limitations of this study section (see above).

Keywords are all methodological. Surely keywords should include area of health care, geographical region, and so on.

We adapted the key words as follows: Process evaluation, realist evaluation, perinatal mortality, capacity development, quality improvement

VERSION 2 – REVIEW

REVIEWER	Pearson, Mark University of Hull, Hull York Medical School
REVIEW RETURNED	18-Mar-2022
GENERAL COMMENTS	In my view, the authors have responded fully to all feedback